# A novel replication initiation region encoded in a widespread *Acinetobacter* plasmid lineage carrying a *bla*NDM-1 gene

Elena Bello-López[1⊙], Ángeles Pérez-Oseguera[1⊙], Walter Santos[2], Miguel Ángel Cevallos[1]*

**1** Centro de Ciencias Genómicas, Programa de Genómica Evolutiva, Universidad Nacional Autónoma de México, Cuernavaca, Morelos, México, **2** Departamento de Microbiología, Instituto de Biotecnología, Universidad Nacional Autónoma de México, Cuernavaca, Morelos, México

⊙ These authors contributed equally to this work.
* mac@ccg.unam.mx

**Data Availability Statement:** The sequence of the plasmid is already deposited in Genbank and the accession number is in the text.

**Funding:** Programa de Apoyo a Proyectos de Investigación e Innovación Tecnológica

## Abstract

The *bla*NDM-1 gene and its variants encode metallo-*beta*-lactamases that confer resistance to almost all beta-lactam antibiotics. Genes encoding *bla*NDM-1 and its variants can be found in several *Acinetobacter* species, and they are usually linked to two different plasmid clades. The plasmids in one of these clades contain a gene encoding a Rep protein of the Rep_3 superfamily. The other clade consists of medium-sized plasmids in which the gene (s) involved in plasmid replication initiation (*rep*)have not yet been identified. In the present study, we identified the minimal replication region of a *bla*NDM-1-carrying plasmid of *Acinetobacter haemolyticus* AN54 (pAhaeAN54e), a member of this second clade. This region of 834 paired bases encodes three small peptides, all of which have roles in plasmid maintenance. The plasmids containing this minimal replication region are closely related; almost all contain *bla*NDM genes, and they are found in multiple *Acinetobacter* species, including *A. baumannii*. None of these plasmids contain an annotated Rep gene, suggesting that their replication relies on the minimal replication region that they share with the plasmid pAhaeAN54e. These observations suggest that this plasmid lineage plays a crucial role in the dissemination of *the bla*NDM-1 gene and its variants.

## Introduction

Carbapenems are broad spectrum b-lactam antibiotics that are used as a last resort in the treatment of critical infections in hospitals. Unfortunately, because of an abrupt increase in carbapenem-resistant Gram-negative pathogenic bacteria, the therapeutic value of these substances is declining rapidly [1, 2]. Carbapenemases that commonly confer carbapenem resistance can be divided into three groups following the Ambler classification: class A and class D are serine-beta-lactamases, and class B are metallo-beta-lactamases [3]. This last group includes the New Delhi metallo-beta-lactamase *bla*NDM-1 and its 60 variants [4–6]. The *bla*NDM-1 gene confers resistance to all beta-lactam antibiotics, except for monobactams and the

(IN204421), Universidad Nacional Autónoma de México. the funder had no role in study design, data collection and analysis, decision to publish, or preparation of the manuscript.

**Competing interests:** The authors have declared that no competing interests exist.

amidinopenicillin mecillinam [7]. The first New Delhi metallo-beta-lactamase [*bla*NDM-1] was discovered in 2008 in a *Klebsiella pneumoniae* strain infecting the urinary tract of a Swedish patient who was returning from New Delhi [8]. Since that time, the *bla*NDM gene and its derivatives have spread rapidly, causing problems in many hospitals. These genes are found in many members of the Enterobacteriaceae family, including *Citrobacter koseri*, *Enterobacter cloacae*, *Escherichia coli*, *K. pneumoniae*, and *Raoultella ornithinolytica*, as well as in species outside the Enterobacteriaceae family, such as *Pseudomonas aeruginosa* and *Acinetobacter baumannii* [9].

The *bla*NDM genes are linked to a wide variety of plasmids, including IncC, IncFII, IncFIB, IncH, IncL/M, and IncX3, and untyped plasmids; however, it always occurs near insertion sequences, (IS5, IS26, ISAba125, IS3000, and ISCR1); transposons, like Tn3, Tn125, and Tn3000, or transposon relics [9–14].

Recently, Tang and coworkers described two large *Acinetobacter* plasmid clades harboring *bla*NDM genes [14]. One of the clades consists of plasmids with a gene encoding a Rep protein of the Rep_3 superfamily. Members of this plasmid clade also carry an untypable relaxase gene belonging to the VirD4/TraG subfamily. The second clade (pNDM-YR7 clade) consists of medium-sized conjugative plasmids (39.36 kb to 49.65 kbthat carry a MOBQ family relaxase gene, encode TrwB/TraD subfamily type IV coupling proteins, and possess VirB-like type IV secretion system gene clusters. Interestingly, no gene encoding a replication initiation protein (Rep) was identified in the members of this second family [14].

The pAhaeAN54e plasmid (NZ_CP041229.1), a member of the pNDM-YR7 clade, is a medium-size replicon that was identified in the *Acinetobacter haemolyticus* strain AN54, isolated in a Mexican hospital. This plasmid contains an NDM-1 gene linked to carbapenem resistance and an *aph*A6 gene involved in aminoglycoside resistance, and both genes lie close to an ISAba125 element [15, 16]. This plasmid also encodes proteins involved in maintenance and transfer and 19 proteins with unknown functions. Our bioinformatics analysis did not reveal any gene encoding a Rep protein for the initiation of plasmid replication on the pAhaeAN54e plasmid or the other members of the pNDM-YR7 clade [14]; in this work, we identified the minimal region of the pAhaeAN54e plasmid necessary for plasmid replication by an experimental approach. This 834 bp region has an unusual organization in that it encodes three small peptides that are required for plasmid replication and maintenance.

## Materials and methods

### Bacterial strains and plasmids

The genomic DNA of *A. haemolyticus* AN54 was used as a PCR template to amplify different regions of one of its plasmids, pAhaeAN54e. The description and genome sequence of this strain have been published previously [15, 16]. To test the replication capabilities of the constructs, a spontaneously pAhaeAN54e-cured strain resistant to rifampicin was used as a recipient for all the constructs. This rifampicin-resistant strain, named AN54De-*rif*, was obtained by plating aliquots from a culture of strain AN54De grown at 37°C and 250 rpm in 5 ml of LB, on LB plates supplemented with 100 μg/ml rifampicin and incubating them at 37°C overnight. A single colony that grew under these conditions was chosen and purified three consecutive times in LB supplemented with 100 μg/ml rifampicin.

The donors in the conjugation experiments were *E. coli* S17.1 derivatives harboring our constructs [17]. *E. coli* DH5α was used to maintain and purify the vectors and constructs [18]. To test the replication capabilities of different pAhaeAN54e DNA fragments, we constructed a shuttle vector incapable of replicating in *A. haemolyticus*. For this purpose, we introduced a gentamycin resistance cassette into the plasmid pBSL142 [19] at the *Mlu*I site of pDO, a ColE1

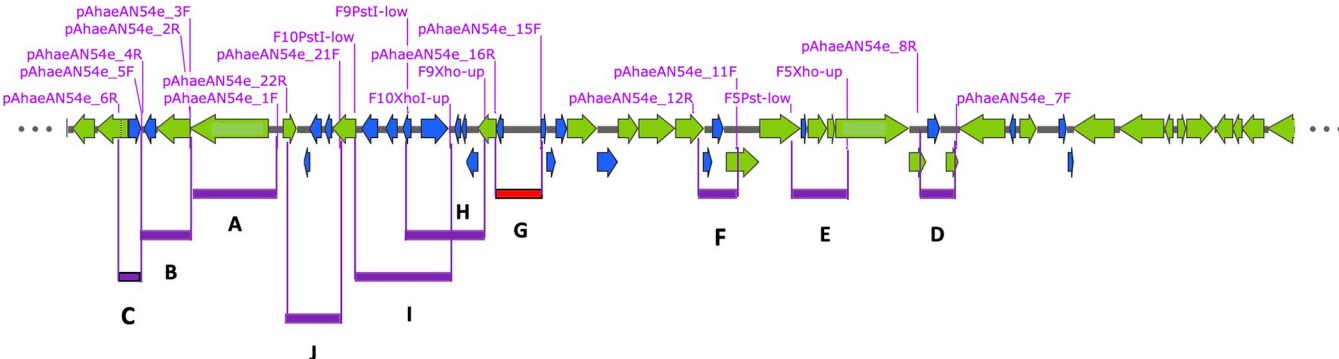

**Fig 1. Linear map of the pAhaeAN54e plasmid.** The arrows in blue indicate genes encoding hypothetical proteins. The arrows in green indicate genes encoding proteins with functional annotations. The red bar filled with the letter "G" shows the fragment containing the minimal replication region of the plasmids. The purple bars indicate the other PCR products that were evaluated for replication function. The positions and names of the PCR primers used in this work are given in magenta.

chloramphenicol-resistance vector [20]. The resultant construct was named pDOG. In all conjugation experiments, the positive control was a gentamycin derivative of the shuttle expression vector pMMB67EH (ATCC number 37623), which is capable of replicating in *Acinetobacter* (donated by Dr. D. Zamorano, CCG, UNAM). Growth curves were obtained in 96-well plates in a Biotek Synergy 2 microplate reader, using LB medium (USA).

## Constructs

Selected regions of the pAhaeAN54e plasmid were amplified via PCR using genomic DNA from the wild-type AN54 strain as the template. The primers used in this process are listed in S1 Table, and their positions are shown in the map in Fig 1. The primers contained unique restriction sites at their 5' ends and were used to insert these restriction sites within the polylinker of the pDOG vector. The PCR products were cut directly with the appropriate restriction enzymes, ligated to the vector cut with the same enzymes, and transformed into *E. coli* DH5α. The sequences of all PCR products were determined by Sanger sequencing. T4 DNA ligase and all the restriction enzymes used in this work were purchased from New England Biolabs (USA). The description of the constructs is presented in Table 1.

## Bacterial mating

**Rapid mating.**   The pDOG constructs containing selected DNA fragments of the plasmid pAhaeAN54e were introduced by transformation into *E. coli* S17-1, using the protocol outlined by [18]. The resultant strains were used as donors to introduce the constructs by conjugation into the AN54Δe-*rif* strain.

Donors and recipients were grown overnight at 37°C and 250 rpm in 3 ml of LB liquid medium without antibiotics. One hundred microliters of recipient culture were plated on MacConkey agar plates supplemented with rifampicin (100 μg/ml) and gentamycin (30 μg/ml). Then, 80 μl of the donor culture was spread in the center of the plate in an approximately 3 cm$^2$ area. The plates were cultivated overnight at 37°C. We considered a cross to be successful when gentamicin- and rifampicin-resistant yellow colonies grew exclusively within the area where the donor strain was plated. In this way, we ruled out spontaneous rifampicin-resistant mutants of the *E. coli* S17.1 donor. The conjugation experiments were performed at least three times. To determine whether the transconjugant strains contained constructs that were

**Table 1. Constructs used in this work.**

| Constructs | GeneBank Coordinates (CP041229.1) | Size (pb) |
|---|---|---|
| pDOG-A | 4,548..7,811 | 3,262 |
| pDOG-B | 2,838..4,584 | 1,746 |
| pDOG-C | 1,904..2,786 | 882 |
| pDOG-D | 31,545..32,985 | 1,440 |
| pDOG-E | 26,870.. 28,933 | 2,063 |
| pDOG-F | 23,227..24,483 | 1,616 |
| pDOG-G | 15,894..17,586 | 1,692 |
| pDOG-H | 12,654..15,510 | 2,856 |
| pDOG-I | 10,671..14,220 | 3,549 |
| pDOG-J | 8,151..10,129 | 1,978 |
| pDOG-BC | 15,894..17,358 | 1,464 |
| pDOG-BCc1 | 15,894..17,212 | 1,318 |
| pDOG-BCc2 | 15,894..17,101 | 1,207 |
| pDOG-BL | 16,194..17,358 | 1,164 |
| pDOG-B215 | 16,270..17,212 | 942 |
| pDOG-B103 | 16,378..17,212 | 834 |

capable of replicating as independent entities, we obtained the plasmid profiles of at least four transconjugants per cross, per experiment, as described below.

**Quantitative mating.** Constructs that successfully replicated in the assay described above were introduced into AN54Δe-*rif* again to calculate their frequency of conjugation. For this purpose, the *E. coli* S17.1 derivatives carrying the selected constructs were grown to the stationary phase in 5 ml of LB medium at 37°C and 250 rpm. The recipient was grown in the same way. Donor and recipient cultures (100 ml each) were mixed on LB plates 1:1 and incubated at 37°C overnight. The cells were resuspended in fresh LB medium, and serial dilutions were plated on MacConkey agar plates supplemented with rifampicin (100 µg/ml) and gentamycin (30 µg/ml). To evaluate the number of recipients, the dilutions were also plated on MacConkey agar plates supplemented with rifampicin (100 µg/ml). The plates were incubated overnight at 37°C. The frequency of conjugation was obtained by calculating the ratio of the number of transconjugants to the number of recipients. The conjugation frequencies were compared using the nonparametric Kolgomorov-Smirnov test implemented in the Python library scipy.

## Mutations

To determine the involvement of the three small coding sequences (CDSs) encoded within the minimal replication region of the plasmid pAhaeAN54e and present in the insert of the construct pDOG-B103, we used overlap extension PCR to construct three mutant derivatives using pDOG-B215 as the PCR template. Mutant derivative pDOG-M2, carried a mutation that eliminated the first codon of the P2 CDS [21]. Construct pDOG-M3, contained a frameshift mutation that affected only the P3 CDS. The construct pDOG-M4 carried a mutation that eliminated the first codon of the P4 CDS.

The primers used in these constructs are listed in **S1 Table**. In general, the mutations were constructed by adding a *Hin*dIII restriction site at the beginning of the reading frame of each small coding sequence. Constructs harboring these new unique restriction sites were digested with *Hin*dIII. Subsequently, the DNA overhangs were eliminated with the "DNA blunting enzyme", which is one of the components of the "CloneJET PCR Cloning Kit" (Thermo Fisher

Scientific, USA). Then, the plasmids were religated with T4 DNA ligase and transformed into *E. coli* DH5a. The correctness of the constructs was evaluated via Sanger sequencing. We tested the functionality of the plasmids carrying the mutations by conjugation using the AN54Δe-*rif* strain as a recipient.

## Plasmid stability

To evaluate the plasmid stability of the vector carrying the minimal replicator region, an AN54Δe-*rif* (pDOG-B103) transconjugant was grown in 20 ml of LB medium supplemented with gentamycin (30 μg/ml) in a 125 ml flask overnight at 37˚C and 250 rpm. Next, the cells were collected by centrifugation, washed twice with fresh LB medium without antibiotics, and inoculated into 25 ml of LB (without antibiotics) in a 125 ml flask at D.O. 620 nm = 0.15. The culture was incubated at 37˚C and 250 rpm. Samples were taken at 0, 3, and 24 h, and dilutions of these samples were plated on LB agar plates and incubated overnight at 37˚C. One hundred colonies per sample were tested on LB agar plates and on LB plates supplemented with genta-mycin (30 μg/ml). The plasmid loss rate was evaluated by counting colonies that remained gentamycin resistant. The experiment was repeated three times. The growth curves were com-pared using the nonparametric Kolgomorov-Smirnov test implemented in the Python library scipy.

## Plasmid profiles

The plasmid profiles of the *A. haemolyticus* AN54Δe-*rif* ⁻transconjugants were visualized on agarose gels according to the protocol described by [22].

## Plasmid sequence analysis

To identify plasmids carrying the minimal replication region of the pAhaeAN54e plasmid, we performed a BLASTn search at NCBI with the default parameters. We selected matches with 100% sequence identity and 100% coverage. The pairwise average nucleotide identity (ANI) based on the MUMmer of the plasmids selected with the criteria mentioned above was calcu-lated using pyANI [23]. The core genome of the plasmids carrying the minimal replication region of the pAhaeAN54e plasmid was obtained with Roary [24]. To identify the open read-ing frames located within the minimal replication region of the pAhaeAN54e plasmid, we used the ExPASy translate tool [Swiss Institute of Bioinformatics, https://web.expasy.org/translate/].

## *In-silico* structural models of the peptides

Structural models of the peptides studied in this work were constructed as follows: A predic-tion round for monomers was performed with the AlphaFold2 (AF2) algorithm [25] on a Linux server with 128 cores, 1 TB of RAM, and an NVIDIA Quadro RTX 6000 GPU through a Jupiter interaction module for Python 3.8 obtained from the DeepMind2 GitHub repository. In brief, the algorithm starts by generating a multiple sequence alignment (MSA) of the prob-lem string with the PDB100 database, then groups the sequence regions into a weight/position matrix and determines the internal protein interactions by comparison with its database. Finally, the last layer treats the resulting structure as a residual gas that moves through the net-work, subject to the mathematical constraints of the algorithm, and performs local and itera-tive refinement to produce the final structure. After the prediction, a search for structural homologs based on secondary structure matching [SSM] was performed with the 3D-blast3

and PDBeFOLD4 algorithms, revealing structures with similarities ranging from 45 to 79% [26, 27].

## Results

*A. haemolyticus* AN54 is a nosocomial strain that was recovered in 2016 from a peritoneal dialysis fluid culture. This strain is resistant to a wide range of beta-lactam antibiotics, with the exception of ticarcillin-clavulanic acid. The strain is also resistant to amikacin but sensitive to gentamycin [15]. Genome sequence analysis indicated that the strain contains five plasmids, one of which, pAhaeAN54e, carries the *bla*NDM-1 gene, which confers resistance to beta-lactam antibiotics. The plasmid size of pAhaeAN54e is 45460 bp, and based on the annotations provided by NCBI, the plasmid encodes 53 CDSs, 19 of which are annotated as hypothetical proteins. The CDSs for these hypothetical proteins are arranged within the plasmid sequence in small clusters (Fig 1). The annotations do not mention any protein involved in plasmid replication. Notably, pAhaeAN54e possesses a *parB* gene, which is involved in plasmid partitioning, but not *parA*, its cognate functional partner.

To identify the gene(s) involved in plasmid replication, we amplified ten regions of the plasmids, some of which encoded hypothetical proteins and some of which were large regions with no annotated CDS. The sizes of these regions ranged from 894 bp to 3,275 bp. The locations of the regions and the positions of the primers used for amplification are shown in Fig 1. We cloned each of the amplified DNA regions into pDOG, a mobilizable shuttle vector that is incapable of replicating in members of the *Acinetobacter* genus but is able to confer gentamycin resistance.

The resultant recombinant plasmids (pDOG-A to pDOG-J) were introduced into the *A. haemolyticus* AN54Δe-*rif* strain so that only the recombinant plasmid carrying the region involved in plasmid replication would be able to establish itself as an independent replication entity within the recipient strain. Importantly, in the process of conjugation, *A. haemolyticus* AN54Δe-*rif* can accept an expression vector with a wide host range that also confers resistant to gentamycin (pMMB67EH). This observation indicated that the AN54Δe-*rif* strain does not have major conjugation barriers that impede the acquisition of new plasmids through conjugation. Moreover, we did not expect problems linked to plasmid incompatibility, considering that AN54Δe-*rif* is a spontaneously cured strain derived by the loss of the plasmid pAhaeAN54e. Only the recombinant plasmid pDOG-G was able to produce transconjugants when the AN54Δe-*rif* strain was used as the recipient (Fig 2). Importantly, we never obtained transconjugants with the empty vector (pDOG).

To determine whether the pDOG-G plasmid replicates as an independent replicon, we obtained the plasmid profiles of the recipient strain as a control and four AN54Δe-*rif* (pDOG-G) transconjugants for each cross. As shown in Fig 3, a new band was observed for the pDOG-G transconjugants but not for the recipient. These results indicated that pDOG-G can replicate as an independent molecule in AN54Δe-*rif*.

### Genes in Fragment G

The insert size of the pDOG-G plasmid is 1692 bp, and the pDOG-G plasmid encompasses the region of the pAhaeAN54e plasmid from position 15894 to 17585 according to the numbering in the GenBank accession of this plasmid. According to the annotations provided by NCBI, the complementary strand of this fragment encodes a small hypothetical protein of 79 aa (NCBI Reference Sequence: WP_005000443.1) and the first five codons of an AAA family ATPase gene. A small hypothetical protein with an identical sequence has been found in other *Acinetobacter* species. However, the NCBI prokaryotic annotation pipeline [28] did not find

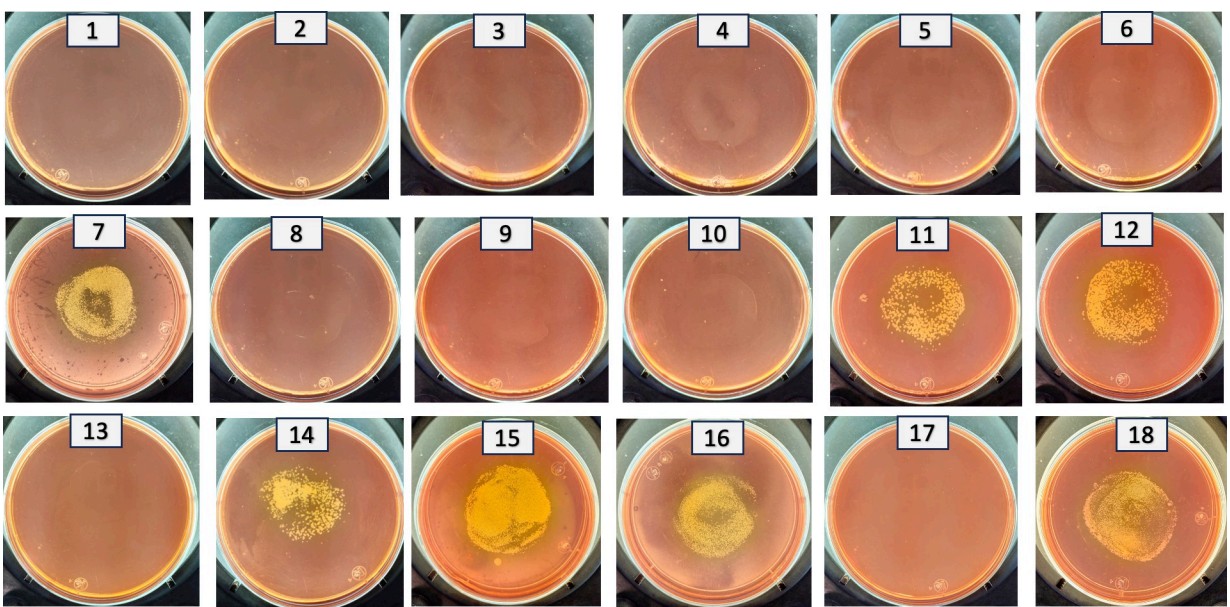

**Fig 2.** All (rapid) genetic crosses used as receptor A. *haemolyticus* AN54Δe-*rif*, and as a donor *E. coli* S.17-1 carrying the following plasmids: 1.-pDOG-A, 2.- pDOG-B, 3.- pDOG-C,4.- pDOG-D, 5.- pDOG-E, 6.- pDOG-F, 7.- pDOG-G, 8.- pDOG-H, 9.- pDOG-I, 10.- pDOG-J, 11.-pDOG-BC, 12.- pDOG-BCc1, 13.- pDOG-BCc2, 14.- pDOG-BL, 15.- pDOG-B215, 16.- pDOG-B103, 17.- pDOG (negative control, empty vector, 18.-pMMB67EH-GM (positive control).Transconjugants appear in the middle of the plates.

any features to annotate from positions 161167 to 17585 (1419 bp). To perform a deeper analysis of this region, we searched for CDSs that were smaller than the minimal cutoff value used by the annotation tool mentioned above. Through this analysis, we found that the insert of pDOG-G potentially encodes four small peptides: the first CDS (positions 15973 to 16081) encodes P1, a 35 aa peptide that partially overlaps with the hypothetical protein mentioned above (WP_005000443.1). The second CDS (positions 16482 to 16682) encodes P2, a 66 aa peptide. The third CDS (positions 16668 to 16812) encodes P3, a 47-aa peptide. The last 14 nucleotides of the 3'-end of the second CDS overlap with the 5' end of the third CDS. The last CDS (positions 17058 to 17202) encodes P4, a 47 aa peptide.

## The minimal replication region

To determine the minimal region involved in the replication of the pAhaeAN54e plasmid and to identify the genes contained in this region, we generated a collection of processive deletions in the 3'-end and 5'-end of Fragment G by PCR. We cloned and inserted these deleted derivatives of Fragment G into pDOG and transferred them to the recipient AN54Δe-*rif* strain by conjugation (Fig 2). A summary of the characteristics and replication abilities of these deletion derivatives is presented in Fig 4. The minimal replication region of the pAhaeAN54e plasmid is contained within the insert in pDOG-B103 (coordinates 16378 to 17212), which contains all the necessary elements to allow the construct to establish itself as an independent replicon in the host strain. The conjugation frequencies of the plasmids pDOG-G, pDOG-B215, and pDOG-B103 fell within the same range, between $2.84 \times 10^{-4}$ and $5.99 \times 10^{-4}$, and statistically indistinguishable (pvalue = 0.1).

The loss rate of pDOG-B103 was relatively low in the absence of selective pressure (Fig 5A). However, the trade-off cost of maintaining pDOG-B103 was greater than the trade-off cost of maintaining pDOG-G. As shown in Fig 5B, strain AN54Δe-*rif* (pDOG-G) showed nearly as much growth as the recipient strain alone.

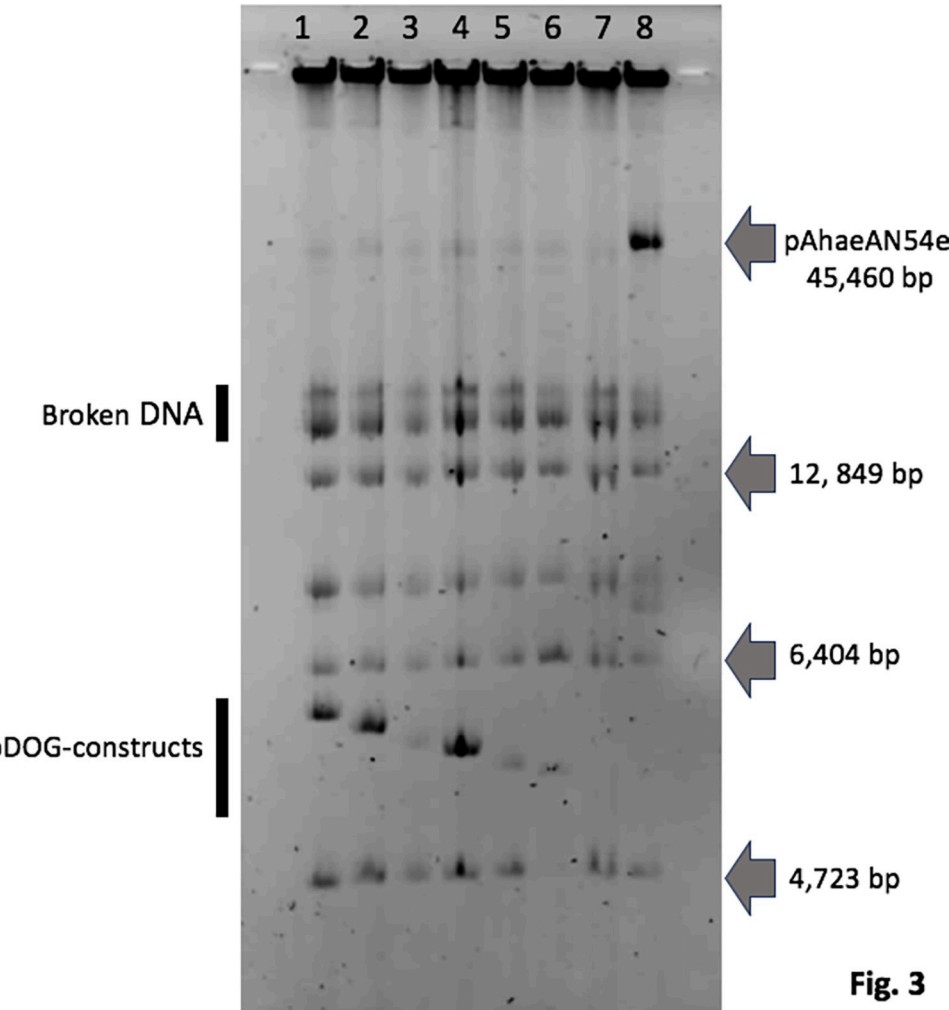

**Fig 3. Plasmid profiles [negative image] of AN54Δe-*rif* transconjugants harboring the following plasmids: 1, pDOG-G; 2, pDOG-BC; 3, pDOG-BCc1; 4, pDOG-BL; 5, pDOG_B215; and 6, pDOG-B103.** Lane 7 shows the plasmid profile of strain AN54Δe-*rif*, and lane 8 shows the plasmid profile of AN54-*rif* (wt). Gray arrows at right indicate the molecular weight of some of the AN54 plasmids.

The pDOG-B103 insert consists of 834 bp, and contains the CDSs of P2, P3, and P4. The insert also contains 103 bp upstream of the initiation codon of the CDS of P1, which is long enough to contain a promoter. Furthermore, the pDOG-B103 insert contains a 10 bp sequence downstream of the stop codon of the P4 CDS. Constructs with deletions that interrupt the CDS of P2 or P4 are incapable of replicating in AN54Δe-*rif*. These results indicate that P4 is essential for replication; however, these observations cannot rule out the necessity of P2 and P3 for two reasons: first, deletions that disrupt the P2 CDS also disrupt the regions downstream; second, the CDSs of P2 and P3 overlap with each other. This structure strongly suggests that P2 and P3 are co-transcribed. However, the results do rule out the participation of the hypothetical protein WP_005000443.1 in replication.

To examine the roles of the peptides in replication, we generated a set of three mutants that modified one CDS without affecting the other two. These constructs, named pDOG-M2, pDOG-M3, and pDOG-M4, were introduced by conjugation into the AN54Δe-*rif* strain, and none of them replicated, indicating that all three peptides are required for replication (Fig 3).

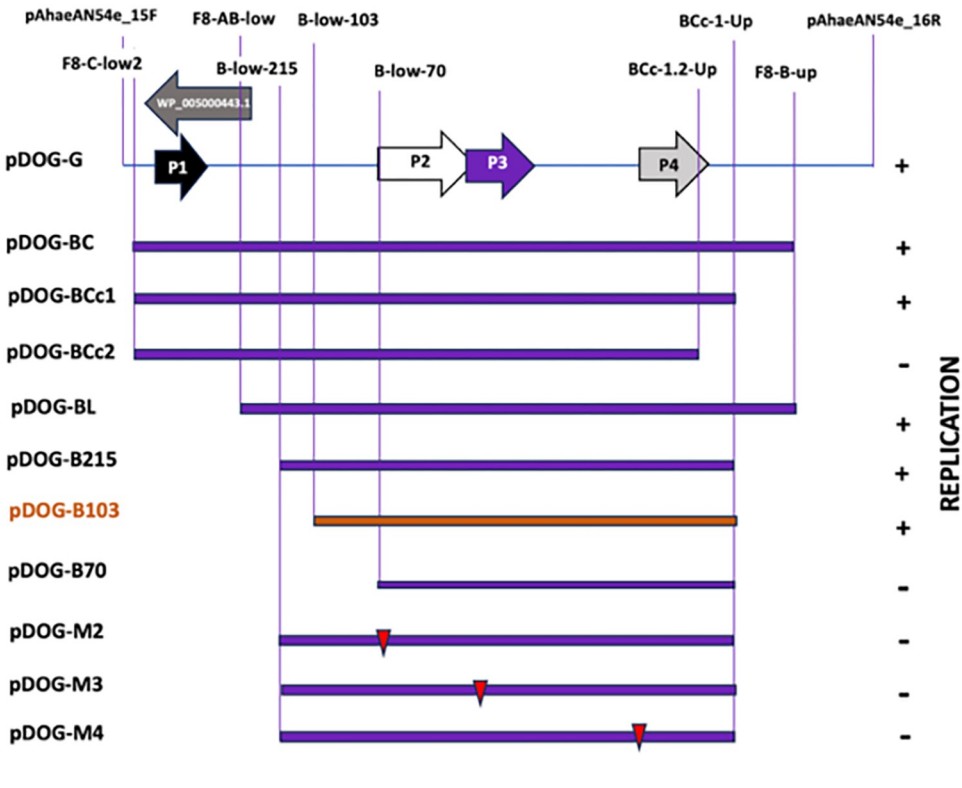

**Fig 4. Map of fragment G indicating the positions of the genes encoding peptides P1, P2, P3, and P4 and of the gene encoding WP_005000443.1.** Below, shown by purple bars, are the different deletion constructs developed in this work. The column on the right shows the replication properties of the constructs (+, capable of replicating); (- incapable of replicating). The names of the constructs are on the left. The smallest region able to sustain plasmid replication (B103) is shown by an orange bar. Red triangles indicate the position of the mutations introduced into the genes encoding peptides P2, P3, and P4. The names of the constructs are given in the left column. The names and positions of the primers used in construction are shown in the upper part of the graphic.

## Host range of the pDOG-B103 plasmid

To explore the host range of the minimal replication region cloned in pDOG-B103, we used two approaches. First, the sequence of the minimal replication region 834 bp was used as a query against the GenBank nr database with BLASTn [29], seeking matches with 100% sequence identity and 100% coverage (August 1, 2023). We found 40 entries that met these requirements. Thirty-seven of the 40 were plasmids from *Acinetobacter* species, including *A. baumannii*, *A. bereziniae*, *A. chinensis*, *A. cumulans*, *A. indicus*, *A. johnsonii*, *A. lactucae*, *A. lwoffii*, and *A. nosocomialis*. However, a plasmid from *Providencia rettgeri* was also a match. Thirty-four of these plasmids, including the *P. rettgeri* p06-1619-NDM plasmid, contained the *bla*NDM gene. The p6200-47.274 kb plasmid did not contain a β-lactamase gene, and p06-1619-NDM rather than containing a *bla*NDM allele, encodes the metallo-beta-lactamase GIM-1 (S2 Table).

The annotations of these plasmids do not mention the presence of Rep proteins. However, two exact matches were also found on the chromosomes of *A. baumannii* WPB103 and *E. coli* WPB121, which suggested that the minimal replication sequence present in the insert of pDOG-B103 can replicate in several *Acinetobacter* species. The *P. rettgeri* p06-1619-NDM plasmid is a different case. This plasmid contains two well-separated modules: one containing all the genes in common with the *Acinetobacter* plasmids listed above and a second module

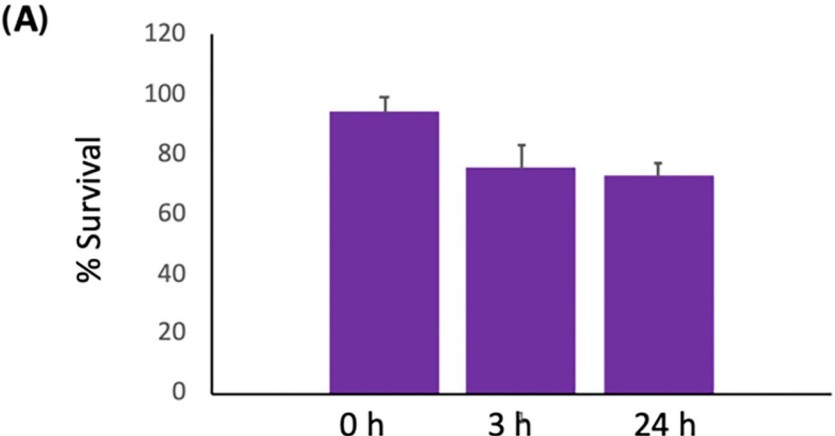

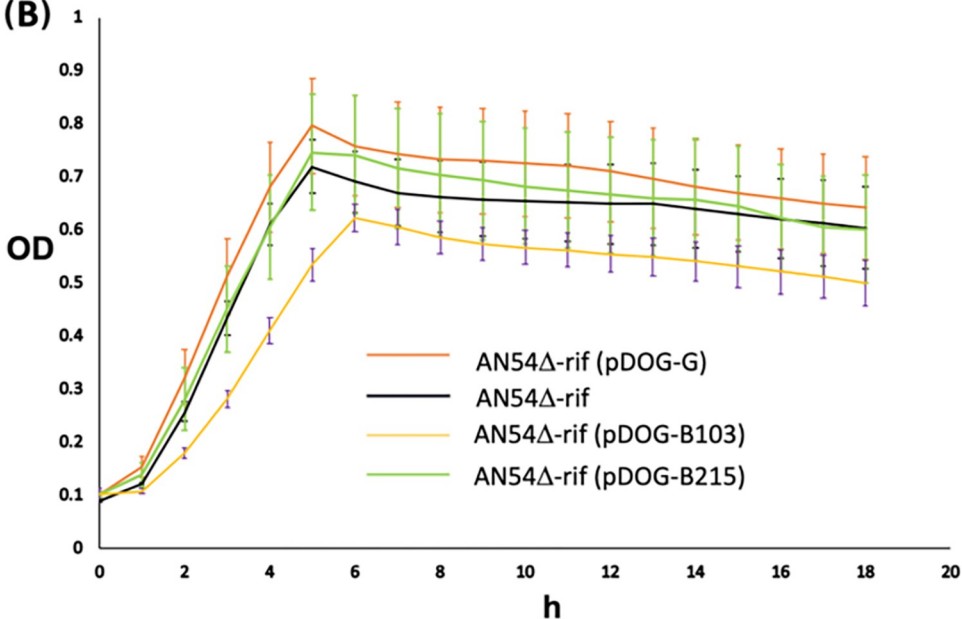

Fig. 5

**Fig 5. A)** Plasmid loss rate of strain AN54Δe-*rif* (pDOG-B123). **B)** Growth curves of strains AN54Δe-*rif*, AN54Δe-*rif* (pDOG-G), AN54Δe-*rif* (pDOG-B215), and AN54Δe-*rif* (pDOG-B123). All the curves showed a beta-like distributions. Samples AN54Δe-*rif*, and AN54Δe-*rif* (pDOG-B215) have the same distribution (pvalue = 0.537). AN54Δe-*rif*, and AN54Δe-*rif* (pDOG-G) have similar distributions (pvalue = 0.026). In contrast, AN54Δe-*rif* (pDOG-B103) deviates from the other three samples by enough significance (p = 0.0001).

similar to those found in other Enterobacteriaceae plasmids. The first module includes the minimal replication region described for pAhaeAN54e. The other contains several genes, including *repA*, an initiator replication-encoding gene [15].

### *In-silico* structural models of the peptides P2, P3, and P4

As shown in **Fig 6**, we modeled the structure of the three peptides *ab initio* with AlphaFold2. P2 and P3 each form a single alpha helix and an unstructured tail at the N-terminus. Although,

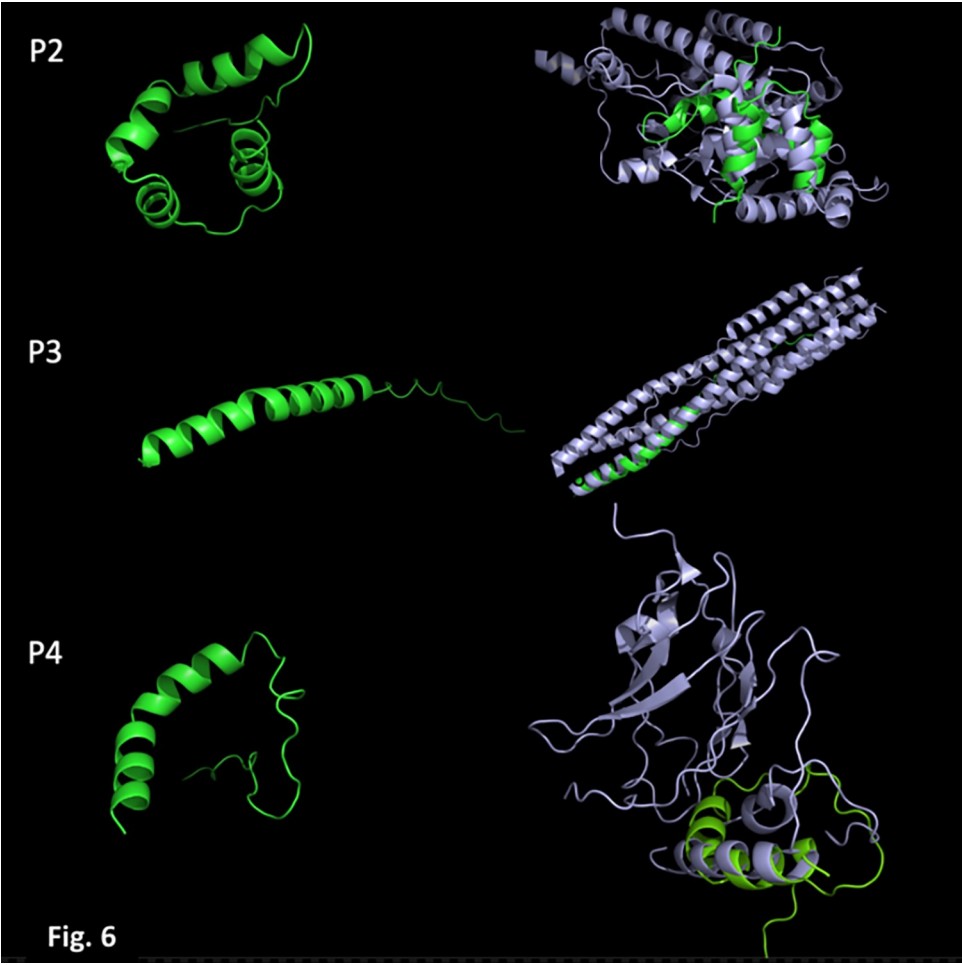

**Fig 6. Structure models of the peptides (in green).** The panels on the left are the structure models of P2, P3, and P4, and on the right, the superimposition of each with the closest 3D match: P2 (Rad4), P3 (HR1HR2 bundle of SARS-CoV-2), and P4 (Filamin A).

P2 also has a small unstructured tail at its N-terminus, instead of a single alpha helix, it contains four small alpha helices linked by loops. A search for structural homologs based on secondary structure matching (SSM) showed that the closest match to P2 was a similar region located within Rad4, a protein involved in the nucleotide excision repair (NER) pathway in yeast. This protein can bind DNA; however, the region in which P2 is structurally similar to Rad4 is not directly involved in DNA–protein interactions (S1 Fig) [30]. The closest match of P3 is located within the post fusion HR1HR2 bundle of the SARS-CoV-2 spike protein [31], and finally, the closest structural match of P4 is a segment of filamin A, a eukaryotic cytoskeleton protein that binds a wide range of proteins [32].

## Discussion

The *bla*NDM-1 gene and its alleles confer resistance to carbapenems, antibiotics that are frequently used to treat severe infections caused by Gram-negative bacteria [33]. These genes are distributed worldwide but are especially prevalent in South Asia, the Balkans, North Africa, and the Middle East. The hosts of these bacteria belong to eleven bacterial families of the class Gammaproteobacteria, including the Moraxellaceae family, which contains the *Acinetobacter*

genus [34]. Interestingly, *bla*NDM genes have been identified in eighteen *Acinetobacter* species, including some that are not linked or are rarely linked to nosocomial infections and a few species that have been isolated from hospital sewage, such as *A. defluvi*, *A. cumulans*, and *A. chinensis* [35]. However, other species have been isolated from the environment; for example, *A. marinus* has been isolated from seawater, *and A. towneri* has been isolated from soil and water [36]. These observations indicate the potential role of *Acinetobacter* in the dissemination of *bla*NDM genes [34]. The *bla*NDM genes are frequently present in plasmids. In a recent work, Tang and coworkers studied a set of 62 *Acinetobacter* plasmids containing *bla*NDM genes [14]. The phylogenetic analysis of these strains revealed that these plasmids belong to two different clades: the pNDM-JN01-like clade and the pNDM-YR7-like clade. The latter consists of plasmids containing conjugative transfer regions similar to those of relaxases in the $MOB_Q$ family, but none of them were found to possess identifiable Rep genes. The pAhaeAN54e plasmid belongs to this group.

A typical plasmid replication region contains a gene encoding an essential rate-limiting initiator replication protein (Rep), an origin of replication, and sequences encoding elements (one or more proteins and/or RNAs) and sites involved in the transcriptional/translational regulation of the *rep* gene [37]. The origins of replication (*oriV*) of plasmids have two discernible regions: one containing the Rep protein binding site and a second containing an AT-rich region at which the origin opens, named the DNA unwinding element (DUE) [38, 39].

The genome of members of the *Acinetobacter* genus consists of one chromosome and frequently one or more plasmids. The Rep proteins of these plasmids belong to several families, which are classified according to their protein domains. The most common Rep proteins belong to the Rep_3 superfamily, followed by Rep proteins belonging to the PriCT-1 superfamily [40]. The pNDM-JN01-like clade of *Acinetobacter* plasmids carrying *bla*NDM genes encodes Rep proteins of the Rep_3 superfamily. However, plasmid lineages in which Rep proteins have not been identified are also common and include the members of the pNDM-YR7--like clade described above [14, 41].

The structure of the minimal replication region of the pAhaeAN54e plasmid is unusual for several reasons: to our knowledge, most *Acinetobacter* plasmids encode only one Rep protein, and their sizes range between 174 aa (i.e., NCBI protein_id: WP_008307689.1) and 432 aa (i.e., NCBI protein_id: WP_059273206.1). In addition, several *Acinetobacter* plasmids encode up to three Rep proteins; however, the genes encoding these proteins are in separate regions of those plasmids, and their origin might be attributable to plasmid fusions [41]. In contrast, the minimal replication region of the pAhaeAN54e plasmid encodes three small peptides (47 aa to 66 aa) clustered in a fragment of 834 bp, and all three peptides are required for plasmid maintenance. Our structural modeling of these peptides did not reveal insights into their functions; however, their 3D structure is similar to segments of other proteins, making it clear that these sequences were not generated by chance or identified by overinterpretation of the bioinformatic analysis that we performed in the search for small CDSs.

Initiator replication (Rep) proteins have at least two essential activities: they recognize and bind to the origin of replication, and they open the replication bubble. The opening of the replication bubble requires the oligomerization of Rep proteins. The peptides encoded in the minimal replication region of the pAhaeAN54e plasmid must therefore possess these properties individually or together. Given the small size of these peptides, we suggest that the peptides need to interact with each other to activate the origin of replication. However, additional experiments are needed to test this hypothesis. Nevertheless, we consider it notable that, to our knowledge, this is the first study in which small peptides were implicated in the initiation and maintenance of plasmid replication.

Another property that is essential for plasmid replication is the presence of an AT-rich region that acts as the DNA unwinding element. The CG content of the minimal replication region was found to be 41.7%, but a section of 102 bp located in the intergenic region between the genes encoding P3 and P4 had a CG content of 27.45% and probably constitutes this DNA unwinding element. Moreover, the minimal replication region does not contain small tandem repeats (iterons), which play a regulatory role in some *Acinetobacter* plasmids [41].

The minimal replication region of the pAhaeAN54e plasmid is not unique. Plasmids containing identical regions are present in at least thirty-eight plasmids carrying *bla*NDM genes and in the same relative positions with respect to their neighboring genes. Most of these plasmids are from distinct *Acinetobacter* species. Moreover, the *P. rettgeri* p06-1619-NDM plasmid also contains this region. Except for p06-1619-NDM plasmid, which contains two replication regions, none of these plasmids contain an annotated Rep gene, suggesting that they replicate using the equivalent genetic elements to those in the minimal replication region of the plasmid pAhaeAN54e.

The general structures of these plasmids are very similar. According to our bidirectional comparisons, 37 of the plasmids show an average nucleotide identity (ANI) higher than 99%, with a coverage of at least 75%. The p06-1619-NDM plasmid exhibits less coverage, between 55% and 62%, but the sequence similarity is still greater than 99%. All these plasmids encode a common set of 28 proteins, of which 20 are hypothetical and one is an aminoglycoside 3'-phosphotransferase. These observations indicate that plasmids that share the minimal replication region of pAhaeAN54e form a well-defined plasmid lineage that plays a crucial role in the dissemination of the *bla*NDM-1 gene and its alleles.

## Supporting information

**S1 Fig.** *In silico* **structural model of the P2 peptide (green) over imposed with Rad4 bound to a mismatch DNA.**
(TIF)

**S1 File.**
(PDF)

**S1 Table. Primers used in this work.**
(PDF)

**S2 Table. Potential Host-range.** Plasmid Blast hits when de DNA sequence of the minimal replication region of plasmid pAhaeAN54e was used as query (100% sequence identity and 100% sequence coverage). Lenght: plasmid sizes. NDM: P (present); ND (not determined), A (absence), GIM (metallo-lactamase GIM-1).
(PDF)

## Acknowledgments

We would like to thank Enrique Merino [IBT-UNAM] and Gabriela Guerrero (CCG_UNAM) for their advice on bioinformatics. Also, to AS Escobedo-Muñoz and Michael Dunn for his critical review of the manuscript.

## Author Contributions

**Conceptualization:** Miguel Ángel Cevallos.

**Data curation:** Elena Bello-López, Ángeles Pérez-Oseguera, Miguel Ángel Cevallos.

**Formal analysis:** Elena Bello-López, Ángeles Pérez-Oseguera, Walter Santos, Miguel Ángel Cevallos.

**Investigation:** Ángeles Pérez-Oseguera, Walter Santos.

**Supervision:** Miguel Ángel Cevallos.

**Writing – original draft:** Miguel Ángel Cevallos.

**Writing – review & editing:** Elena Bello-López, Ángeles Pérez-Oseguera, Walter Santos, Miguel Ángel Cevallos.

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
