## [Decision Letter · Decision Letter 0]

5 Apr 2024

PONE-D-24-08483A novel replication initiation region encoded in a widespread Acinetobacter plasmid lineage carrying a blaNDM-1 genePLOS ONE

Dear Dr. Cevallos,

Thank you for submitting your manuscript to PLOS ONE. After careful consideration, we feel that it has merit but does not fully meet PLOS ONE’s publication criteria as it currently stands. Therefore, we invite you to submit a revised version of the manuscript that addresses the points raised during the review process.

We look forward to receiving your revised manuscript.

Kind regards,

Feng Gao

Academic Editor

PLOS ONE

“Programa de Apoyo a Proyectos de Investigación e Innovación Tecnológica (IN204421), Universidad Nacional Autónoma de México”

Reviewers' comments:

Reviewer's Responses to Questions

**Comments to the Author**

1. Is the manuscript technically sound, and do the data support the conclusions?

Reviewer #1: Yes

Reviewer #2: Yes

2. Has the statistical analysis been performed appropriately and rigorously? 

Reviewer #1: N/A

Reviewer #2: No

3. Have the authors made all data underlying the findings in their manuscript fully available?

Reviewer #1: Yes

Reviewer #2: Yes

4. Is the manuscript presented in an intelligible fashion and written in standard English?

Reviewer #1: Yes

Reviewer #2: Yes

5. Review Comments to the Author

Reviewer #1: The manuscript entitled "A novel replication initiation region encoded in a widespread Acinetobacter plasmid lineage carrying a blaNDM-1 gene" is well written and includes an interesting work therefore, I only have very minor comments:

- line 78: the sentence "always occurs near transposons, such as IS5, IS26, ISAba125, IS3000, ISCR1, Tn3, Tn125, and Tn3000". Amongst those listed IS5, IS26, ISAba125, IS3000, ISCR1 are not transposons, they are insertion sequences. Please modify the sentence.

- Line 219: Change this heading to "in-silico Structural models of the peptides"

Line 239: Please include the GenBank accession number for pAhaeAN54e.

I'd replace Figure 1 with a more advanced Figure.

Reviewer #2: The article by Bello-Lopez et al presents some interesting findings around the identification of a region in some Acinetobacter plasmids that is required for replication. Until now these plasmids were not known to contain a replication initiation region, and so this study furthers our knowledge of the biology of this group of plasmids. The study is overall well written and has been conducted thoroughly. There are a few points that I think the authors should address:

1. I did not notice any statistical analyses of data such as differences (or not) in conjugation frequencies, rate of plasmid loss, and growth rates with different constructs. These should be carried out to determine whether any differences reach statistical significance, and these data displayed on the appropriate figures e.g. Figure 5A and B.

2. I didn't understand Figure 2. From the methods section I understood that if a collection of yellow colonies grew in the center of the plate, then these represented the transconjugants. In the results (lines 264-266) it refers to figure 2 and days that only pDOG-G was able to produce transconjugants, but Figure 2 appears to show transconjugants for pDOG-G, pDOG-B215 and pDOG-B103. Please can further explanation be provided to explain this. If the authors have the pictures, it might be worth showing the plates for the other constructs that didn't produce transconjugants as well, rather than just the positive ones.

3. The legend for Figure 3 refers to pDOG-BCc, but elsewhere in the manuscript there are two plasmids called pDOG-BCc1 and pDOG-BCc2. Which is being referred to in Figure 3? Further, the legend refers to a plasmid called pDOG-B123 - should this be pDOG-B103?

4. In places, mainly in the introduction, the language needs to be tightened up a bit as it is sometimes inconsistent, misleading, or erroneous. Examples include:

Line 59: change 'resource' to 'resort'.

Line 64: be consistent with how you write the 'beta' for beta-lactamases.

It would be more accurate to refer to the gene as blaNDM, and the allele as blaNDM-1 (so the blaNDM gene and its variants).

Line 68: I think this is the first mention of K. pneumoniae so Klebsiella should be written fully.

Line 70: the phrase 'causing problems in every hospital they have reached' is a bit over the top, and not something we can support with evidence.

Lines 77-78: you refer to transposons and then list some insertion sequences.

Line 372-373: some italicising and bolding has over-run.

6. PLOS authors have the option to publish the peer review history of their article (what does this mean?). If published, this will include your full peer review and any attached files.

Reviewer #1: No

Reviewer #2: **Yes: **Benjamin A Evans

---

## [Author Response · Author response to Decision Letter 0]

18 Apr 2024

ANSWER TO REVIEWERS

5. Review Comments to the Authors

Reviewer #1: The manuscript entitled "A novel replication initiation region encoded in a widespread Acinetobacter plasmid lineage carrying a blaNDM-1 gene" is well written and includes an interesting work therefore, I only have very minor comments: 

Line 78: the sentence "always occurs near transposons, such as IS5, IS26, ISAba125, IS3000, ISCR1, Tn3, Tn125, and Tn3000". Amongst those listed IS5, IS26, ISAba125, IS3000, ISCR1 are not transposons, they are insertion sequences. Please modify the sentence.

Line 219: Change this heading to "in-silico Structural models of the peptides

Answer: Changed as suggested.

Line 239: Please include the GenBank accession number for pAhaeAN54e.

Answer: Included as suggested.

I'd replace Figure 1 with a more advanced Figure.

 Reviewer #2: The article by Bello-Lopez et al presents some interesting findings around the identification of a region in some Acinetobacter plasmids that is required for replication. Until now these plasmids were not known to contain a replication initiation region, and so this study furthers our knowledge of the biology of this group of plasmids. The study is overall well written and has been conducted thoroughly. There are a few points that I think the authors should address:

1. I did not notice any statistical analyses of data such as differences (or not) in conjugation frequencies, rate of plasmid loss, and growth rates with different constructs. These should be carried out to determine whether any differences reach statistical significance, and these data displayed on the appropriate figures e.g. Figure 5A and B.

Answer: statistics now included in the figure legend.

2. I didn't understand Figure 2. From the methods section I understood that if a collection of yellow colonies grew in the center of the plate, then these represented the transconjugants. In the results (lines 264-266) it refers to figure 2 and days that only pDOG-G was able to produce transconjugants, but Figure 2 appears to show transconjugants for pDOG-G, pDOG-B215 and pDOG-B103. Please can further explanation be provided to explain this. If the authors have the pictures, it might be worth showing the plates for the other constructs that didn't produce transconjugants as well, rather than just the positive ones.

Answer: The new figure 2 include all constructs

3. The legend for Figure 3 refers to pDOG-BCc, but elsewhere in the manuscript there are two plasmids called pDOG-BCc1 and pDOG-BCc2. Which is being referred to in Figure 3? Further, the legend refers to a plasmid called pDOG-B123 - should this be pDOG-B103?

Answer: corrections revised and included in the new text.

4. In places, mainly in the introduction, the language needs to be tightened up a bit as it is sometimes inconsistent, misleading, or erroneous. Examples include:

Line 59: change 'resource' to 'resort'.

Answer: correction included in the new text.

Line 64: be consistent with how you write the 'beta' for beta-lactamases.

It would be more accurate to refer to the gene as blaNDM, and the allele as blaNDM-1 (so the blaNDM gene and its variants).

Answer: in the new we homogenized the nomenclature of beta-lactamases.

Line 68: I think this is the first mention of K. pneumoniae so Klebsiella should be written fully.

Answer: correction made as suggested

Line 70: the phrase 'causing problems in every hospital they have reached' is a bit over the top, and not something we can support with evidence.

Answer: this phrase was eliminated in the new text

Lines 77-78: you refer to transposons and then list some insertion sequences.

Answer: correction made as suggested.

Line 372-373: some italicising and bolding has over-run.

Answer: correction made.

---

## [Decision Letter · Decision Letter 1]

6 May 2024

A novel replication initiation region encoded in a widespread Acinetobacter plasmid lineage carrying a blaNDM-1 gene

PONE-D-24-08483R1

Dear Dr. Cevallos,

We’re pleased to inform you that your manuscript has been judged scientifically suitable for publication and will be formally accepted for publication once it meets all outstanding technical requirements.

Kind regards,

Feng Gao

Academic Editor

PLOS ONE

Additional Editor Comments (optional):

Reviewers' comments:

Reviewer's Responses to Questions

**Comments to the Author**

1. If the authors have adequately addressed your comments raised in a previous round of review and you feel that this manuscript is now acceptable for publication, you may indicate that here to bypass the “Comments to the Author” section, enter your conflict of interest statement in the “Confidential to Editor” section, and submit your "Accept" recommendation.

Reviewer #1: All comments have been addressed

2. Is the manuscript technically sound, and do the data support the conclusions?

Reviewer #1: Yes

3. Has the statistical analysis been performed appropriately and rigorously? 

Reviewer #1: Yes

4. Have the authors made all data underlying the findings in their manuscript fully available?

Reviewer #1: Yes

5. Is the manuscript presented in an intelligible fashion and written in standard English?

Reviewer #1: Yes

6. Review Comments to the Author

Reviewer #1: in insertion sequences "IS" and transposon "Tn" names, numbers must be italic. Please fix throughout!

7. PLOS authors have the option to publish the peer review history of their article (what does this mean?). If published, this will include your full peer review and any attached files.

Reviewer #1: No

---

## [Editor Report · Acceptance letter]

21 May 2024

PONE-D-24-08483R1 

PLOS ONE

Dear Dr. Cevallos, 

I'm pleased to inform you that your manuscript has been deemed suitable for publication in PLOS ONE. Congratulations! Your manuscript is now being handed over to our production team.

Kind regards, 

on behalf of

Dr. Feng Gao 

Academic Editor

PLOS ONE